# Permeation of Molecules through Astroglial Connexin 43 Hemichannels Is Modulated by Cytokines with Parameters Depending on the Permeant Species

**DOI:** 10.3390/ijms21113970

**Published:** 2020-06-01

**Authors:** Juan C. Sáez, Aníbal A. Vargas, Diego E. Hernández, Fernando C. Ortiz, Christian Giaume, Juan A. Orellana

**Affiliations:** 1Departamento de Fisiología, Pontificia Universidad Católica de Chile, Santiago 8330024, Chile; jsaez@bio.puc.cl (J.C.S.); dehernan@uc.cl (D.E.H.); 2Instituto de Neurociencias, Centro Interdisciplinario de Neurociencias de Valparaíso, Universidad de Valparaíso, Valparaíso 2381850, Chile; 3Instituto de Ciencias de la Salud, Universidad de O′Higgins, Rancagua 2820000, Chile; anibal.vargas@uoh.cl; 4Mechanisms on Myelin Formation and Repair Laboratory, Instituto de Ciencias Biomédicas, Facultad de Ciencias de la Salud, Universidad Autónoma de Chile, Santiago 8910060, Chile; fernando.ortiz@uautonoma.cl; 5Center for Interdisciplinary Research in Biology (CIRB), College de France, CNRS, INSERM, PSL Research University, 75005 Paris, France; christian.giaume@college-de-france.fr; 6Departamento de Neurología, Escuela de Medicina and Centro Interdisciplinario de Neurociencias, Facultad de Medicina, Pontificia Universidad Católica de Chile, Santiago 8330024, Chile

**Keywords:** connexons, dyes, permeability tracers, permeation, TNF-α, IL-1β, astrocytes

## Abstract

Recent studies indicate that connexin hemichannels do not act as freely permeable non-selective pores, but they select permeants in an isoform-specific manner with cooperative, competitive and saturable kinetics. The aim of this study was to investigate whether the treatment with a mixture of IL-1β plus TNF-α, a well-known pro-inflammatory condition that activates astroglial connexin 43 (Cx43) hemichannels, could alter their permeability to molecules. We found that IL-1β plus TNF-α left-shifted the dye uptake rate vs. dye concentration relationship for Etd and 2-NBDG, but the opposite took place for DAPI or YO-PRO-1, whereas no alterations were observed for Prd. The latter modifications were accompanied of changes in K_d_ (Etd, DAPI, YO-PRO-1 or 2-NBDG) and Hill coefficients (Etd and YO-PRO-1), but not in alterations of V_max_. We speculate that IL-1β plus TNF-α may distinctively affect the binding sites to permeants in astroglial Cx43 hemichannels rather than their number in the cell surface. Alternatively, IL-1β plus TNF-α could induce the production of endogenous permeants that may favor or compete for in the pore-lining residues of Cx43 hemichannels. Future studies shall elucidate whether the differential ionic/molecule permeation of Cx43 hemichannels in astrocytes could impact their communication with neurons in the normal and inflamed nervous system.

## 1. Introduction

The traditional notion of neurons acting as the unique functional unit at the synaptic cleft has been challenged with the discovery that intra and intercellular Ca^2+^ waves within and among astrocytes, respectively, underpin the regenerative (non-dissipative) transfer of physiological signals [1,2,3]. Despite astrocytes not being electrically silent cells [4], changes in intracellular free-Ca^2+^ concentration ([Ca^2+^]_i_ underlie the Ca^2+^ waves that represent a relevant time-scale mechanism for allowing rapid intra and intercellular signaling [5]. Equipped with these tools and accompanying the pre and postsynaptic neurons, astrocytes embrace the “tripartite synapse”—the centerpiece of the chemical synaptic transmission—where they sense neural function and react to it by the [Ca^2+^]_i_-dependent release of biomolecules that modulate neuronal activity called “gliotransmitters” [6]. In recent decades, connexin hemichannels have arisen as essential elements of a new mechanism involved in the astrocyte-dependent release of chemical signals, called gliotransmission [7].

Connexins, the building blocks of hemichannels, embrace a highly conserved protein family encoded by 21 genes in humans and 20 in mice, with orthologs in other vertebrate species [8]. When six connexins oligomerize around a central pore, they form a hemichannel or connexon that allow cellular communication through two mechanisms. At one end, they may dock with hemichannels of neighboring cells at cell–cell interfaces, forming aggregates of intercellular channels that permit the direct but selective molecular and ionic exchange between the cytoplasm of contacting cells [9]. On the other hand, the presence of functional and solitary hemichannels in “non-junctional” membranes may serve as a diffusional route for the release of relevant quantities of autocrine and paracrine signaling molecules (e.g., ATP, glutamate, D-serine, NAD^+^ and PGE_2_) and metabolites (i.e., GSH) as well as the influx of cell signals (i.e., Ca^2+^ and cADPR) and metabolites (i.e., glucose) [10].

Connexin 43 (Cx43) is the most abundant connexin expressed by astrocytes in the brain [11]. Cellular signaling and gliotransmitter release linked to the function of astrocytic hemichannels sustain crucial biological processes at the nervous system, such as neuronal oscillations [12], astroglial migration [13], synaptic transmission and plasticity [14,15,16], as well as memory consolidation and behavior [17,18]. Nevertheless, the increase in open probability of astroglial Cx43 hemichannels has been implicated with the pathogenesis and progression of homeostatic imbalance in various neuropathological diseases [19,20,21,22,23,24,25]. The latter has led to the thought that connexin hemichannels may undergo alterations in their permeability properties during pathological conditions [7].

Previous studies have shown that cation permeation through Cx43 hemichannels is cooperative, competitive and saturable with parameters depending on the permeant species [26]. More relevantly, connexin hemichannels seem to exhibit differential permeabilities towards ions, dyes and small biologically relevant molecules. In particular, recent findings suggest that open connexin hemichannels do not act as freely permeable non-selective pores, but they select permeants in an isoform-specific form, suggesting that fluorescent dye uptake cannot be always employed as a reliable indicator of current conductance or permeation to small biologically relevant molecules [27,28,29,30]. With this in mind, the aim of this study was to investigate whether treatment with a mixture of TNF-α plus IL-1β alters the permeability of astrocytic Cx43 hemichannels to small molecules.

## 2. Results

### 2.1. Cx43 Hemichannel-Dependent Uptake of Permeability Probes Is Increased by Removal of Divalent Cations in Cultured Astrocytes

Although, in vivo, astrocytes express abundant amounts of Cx43 and Cx30 [31], and some of them express low amounts of Cx26 [32], highly enriched cortical astrocyte cultures express only Cx43 [11,33]. This protein forms functional hemichannels in normal and reactive astrocytes as described using different in vitro and ex vivo experimental approaches [14,15,21,23,24]. Because physiological concentrations of extracellular Ca^2+^ and Mg^2+^ reduce the open probability of Cx43 hemichannels [26,34,35], we used a divalent cation-free solution (DCFS) to explore the permeability properties of these channels in cultured astrocytes. Dye uptake or leakage methods have been widely used to approach the functional state of undocked hemichannels in diverse cellular systems [36]. In particular, dyes that become fluorescent upon their binding to nucleic acids, such as ethidium (Etd) or YO-PRO-1, provide a powerful experimental alternative to quantify, in time-lapse imaging, the hemichannel permeability in a non-invasive manner [36,37]. The exogenous expression of Cx43-EGFP hemichannels in HeLa cells enables the cellular uptake of diverse cationic dyes that strongly correlate with the expression of Cx43-EGFP either in the presence or absence of extracellular Ca^2+^/Mg^2+^ [26,34,37]. With the above in mind, we first explored whether astroglial Cx43 hemichannels could show similar permeability properties for the following cationic permeability tracers: Etd, propidium (Prd), DAPI and YO-PRO-1. We used subconfluent and low-density cultures containing relatively isolated astrocytes in order to reduce the possible interference of traffic and/or function of Cx43 gap junction channels on the activity of Cx43 hemichannels.

DCFS increased by ~3-, ~7.5-, ~2.7- and ~3.5-fold the uptake of Etd, Prd, DAPI and YO-PRO-1, respectively (Figure 1A–L). With or without the removal of Ca^2+^/Mg^2+^ from the bath solution, dye uptake was directly correlated with the total amounts of Cx43 for all permeability tracers studied (Figure 1I–L). We observed a direct association between total and cell surface amount of Cx43 (Figure 2A–D), whereas the DCFS-induced uptake of all tracers studied strongly correlated with the fluorescence intensity of Cx43 located at unopposed membranes (Figure 2E–H). Because these data suggested that Cx43 hemichannels participate in the uptake of cationic dyes in cultured astrocytes, we decided to obtain further support to this interpretation using a pharmacological approach to scrutinize the potential contribution of Cx43 hemichannels in the DCFS-induced uptake of the permeability tracers used. For that purpose, astrocytes were pre-incubated for 15 min before and throughout dye uptake experiments with 100 µM gap19 or 100 µM Tat-L2, two selective mimetic peptides that rapidly decrease Cx43 hemichannel activity [38,39,40]. Gap19 or Tat-L2 greatly reduced the DFCS-induced uptake of Etd, Prd, DAPI or YO-PRO-1 to basal values (Figure 2I–T). Given that the inhibitory actions of gap19 and Tat-L2 rely on their homology to the intracellular L2 loop regions of Cx43, we employed a mutated peptide, Tat-L2^H126K/I130N^, in which two amino acids essential for the interaction of the L2 region with the carboxyl tail of Cx43 are modified, as well as an inactive form of gap19 containing the I130A variation (gap19^I130A^). Both gap19^I130A^ or Tat-L2^H126K/I130N^ failed in reducing the DFCS-induced uptake of Etd, Prd, DAPI or YO-PRO-1, suggesting that Cx43 hemichannels are crucial for this response in cultured astrocytes (Figure 2I–L). This idea was reaffirmed by the absence of DFCS-induced dye uptake in astrocytes cultured from Cx43^−/−^ mice (Figure 2I–L).

### 2.2. IL-1β Plus TNF-α Modulates the Permeation of Permeability Tracers through Astroglial Cx43 Hemichannels Depending on the Permeant Species

Treatment with IL-1β/TNF-α has been proved to increase Cx43 hemichannel activity in diverse cell types including astrocytes, myofibers, retinal epithelium and endothelial cells [19,41,42,43,44]. In cultured astrocytes, 24 h treatment with a mixture of these two cytokines (10 ng/mL for each) enhances Etd uptake and current unitary events linked to the opening of Cx43 hemichannels [42]. In this context, we examined whether IL-1β/TNF-α could modify the cell membrane permeability for cationic dyes under basal or DFCS conditions in cultured astrocytes. As expected, the treatment with IL-1β/TNF-α for 24 h induced ~3-fold increase in Etd uptake in relation to control conditions, this response was enhanced in the absence of Ca^2+^/Mg^2+^ in the bath solution (Figure 3A,B,I). The IL-1β/TNF-α-induced Etd uptake recorded in DCFS was completely reduced by either gap19 (100 µM) or Tat-L2 (100 µM), but neither affected by gap19^I130A^ (100 µM) nor Tat-L2^H126K/I130N^ (100 µM) (Figure 3I). Unlike to what was observed with Etd, the uptake of Prd did not change upon treatment, with IL-1β/TNF-α being similar before and after removal of Ca^2+^/Mg^2+^ from the bath solution (Figure 3C,D,J). Noteworthy, astrocytes stimulated with IL-1β/TNF-α showed ~65% or ~50% reduction in DAPI or YO-PRO-1 uptake, respectively, when compared to control astrocytes (Figure 3E–H,K,L). Despite the above, the absence of extracellular Ca^2+^/Mg^2+^ in the bath solution increased DAPI or YO-PRO-1 uptake in astrocytes treated with IL-1β/TNF-α (Figure 3K,L). It should be noted that gap19 or Tat-L2, but not gap19^I130A^ or Tat-L2^H126K/I130N^ totally blunted the uptake of all dyes studied in control or IL-1β/TNF-α-stimulated astrocytes bathed with DCFS (Figure 3I–L).

Several years ago, we demonstrated that Cx43 hemichannels exposed to DCFS permit the positive cooperative permeation of low concentrations of Etd, Prd and DAPI and saturation at higher concentrations [26]. Accordingly, we further scrutinize whether astroglial Cx43 hemichannels could show equivalent permeability features and if IL-1β/TNF-α may modulate their permeability features. Dye uptake kinetics were higher as greater was the concentration of the dyes used in control astrocytes bathed with DCFS (Figure 3M–P). In addition, the relationships between dye uptake rate and concentration for all dyes studied were positive (Figure 3Q–T), and the maximal rates were reached at concentrations of ~150 µM for Etd (Figure 3Q), ~100 µM for Prd (Figure 3R), ~100 µM for DAPI (Figure 3S) or ~50 µM for YO-PRO-1 (Figure 3T). Under these conditions, the ranking of maximal uptake capacity (V_max_) of astroglial Cx43 hemichannels was: Etd > DAPI > YO-PRO-1 > Prd (V_max_ = 14.8 ± 1.2; 6.2 ± 0.3; 1.9 ± 0.2 and 1.4 ± 0.1 AU/min, respectively) (Table 1). Inspection of the uptake rate curves in relation to different dye concentrations reveals a sigmoid pattern, suggesting potential cooperative interactions between permeant molecules and astroglial Cx43 hemichannels at low tracer concentrations (Figure 2E–H). The latter harmonizes with the fact that Hill coefficients were all positive and >1 for Etd (3.2 ± 0.7), Prd (3.1 ± 0.8), DAPI (2.6 ± 0.7) and YO-PRO-1 (1.4 ± 0.4), while the apparent affinity constants (K_d_) were 80.9 ± 8.0 µM, 54.3 ± 5.3 µM, 39.5 ± 4.2 µM and 22.5 ± 5.5 µM, respectively (Table 1).

Noteworthy, the treatment with IL-1β/TNF-α left- or right-shifted the dye concentration response curves depending on the permeant species (Figure 3Q–T). In particular, when compared to control conditions, IL-1β/TNF-α-stimulated astrocytes exhibited maximal rates of uptake at lower concentrations for the case of Etd (~100 µM) (Figure 3Q) but the opposite occurred for DAPI (~150 µM) (Figure 3S) or YO-PRO-1 (~150 µM) (Figure 3T), whereas no changes were detected for Prd (~100 µM) (Figure 3R). Although the V_max_ of astroglial Cx43 hemichannels for Etd, Prd, DAPI or YO-PRO-1 remained unaltered between control or IL-1β/TNF-α-astrocytes, the Hill coefficients and K_d_ were modulated upon treatment with cytokines depending on the permeant species (Table 1). Specifically, IL-1β/TNF-α did not alter the Hill coefficient for DAPI or Prd, but increased it for YO-PRO-1 (from 1.39 ± 0.3 to 2.9 ± 0.7), whereas the opposite took place for Etd (from 3.2 ± 0.7 to 2.1 ± 0.5) (Table 1). Moreover, the K_d_ for DAPI and YO-PRO-1 were increased upon treatment with IL-1β/TNF-α (from 39.4 ± 4.2 to 88.7 ± 11.1 µM and 22.4 ± 5.5 to 82.9 ± 11.1 µM, respectively), whereas a reduction was found for Etd (from 80.9 ± 8 to 43.5 ± 6 µM) (Table 1). Ethanol (70%) was used to induce membrane permeabilization of astrocytes and see whether saturation of the nucleic acid binding sites might contribute to the cooperative (S shape) dye uptake observed in our study (Figure 3Q–T). Given that 70% ethanol caused a much-pronounced increase in dye uptake rate to levels far above the maximal values observed with unpermeabilized cells, the number of intracellular binding sites likely did not affect the kinetics or saturating uptake rate (Figure 4A). Similar findings have been seen in Cx43 hemichannels expressed in HeLa cells [26]. Congruent with the above, the relation between dye uptake and concentration of Etd, Prd, DAPI or YO-PRO-1 in cultured astrocytes permeabilized with ethanol was linear, which imply an unrestricted passive diffusion rather than restricted permeation with a saturable component (V_max_), as described here for Cx43 hemichannels (Figure 4B). Although the dye uptake rate could be affected by the number of hemichannels in the plasma membrane, it has been shown that cells bathed with DCFS do not exhibit changes in the amount of surface Cx43, as measured by the biotinylation of membrane surface proteins [37]. Despite that, we conducted experiments in which dye uptake was normalized to the Cx43 fluorescence at cellular contour in order to minimize variations due to differences in the number of hemichannels per astrocyte. We found that major changes evoked by DCFS and/or IL-1β/TNF-α on astroglial dye uptake (Figure 2I–L and Figure 3Q–S) remained almost unaltered for all dyes studied after normalizing the data to the amount of Cx43 (Figure 4C).

### 2.3. IL-1β Plus TNF-α Does Not Affect the Competitive Permeation of Fluorescent Dyes via Cx43 Hemichannels in Cultured Astrocytes

Studies in HeLa cells transfected with Cx43-EGFP have revealed that Cx43 hemichannels permit the simultaneous passage of more than one fluorescent dye through a competitive process [26]. To explore whether astroglial Cx43 hemichannels behave similarly, we employed Etd and DAPI, which emit fluorescence at different wave lengths and thereby, the presence of one does not perturb the recording of the other molecule. Accordingly, control astrocytes were exposed to different concentrations of Etd or DAPI in the presence of 150 µM DAPI or Etd (Figure 5A–F). The latter dye concentration is within the range that competitively reduce the uptake of Etd or DAPI via Cx43-EGFP hemichannels [26]. In harmony with this evidence, we found that DAPI right-shifted the sigmoidal curve of Etd uptake without changing the V_max_ or Hill coefficient, but increasing the K_d_ (from 80.9 ± 8.0 µM to 133.1 ± 17.8 µM) (Figure 5G). Similar responses were found in IL-1β/TNF-α-treated astrocytes, where 150 µM DAPI also induced the left-shifting of the Etd uptake curve along with a rise in the K_d_ for Etd (from 43.6 ± 6.0 µM to 94.3 ± 19.1 µM) (Figure 5G).

On the other hand, 150 µM Etd caused a right-shift in the DAPI uptake without changing the V_max_ or Hill coefficient, but increasing the K_d_ for DAPI (from 39.5 ± 4.2 µM to 102.3 ± 16.9 µM) (Figure 5H). Likewise, 150 µM Etd did not alter the V_max_ or Hill coefficient in IL-1β/TNF-α-treated astrocytes, but right-shifted the DAPI uptake curve and increased the K_d_ for DAPI (from 88.75 ± 11.1 to 128.1 ± 25.7 µM) (Figure 5H). We next determined the potency of the competitive inhibition of DAPI on Etd uptake rate and vice versa. DAPI induced a concentration-dependent inhibition of Etd uptake rates in both control and IL-1β/TNF-α-treated astrocytes, with an IC_50_ of 108.8 ± 28.2 and 168.9 ± 24.2 µM, respectively (Figure 5I). In the same line, Etd caused a strong concentration-dependent inhibition of DAPI uptake rates with an IC_50_ of 84.6 ± 13.6 and 20.1 ± 4.3 µM for control and IL-1β/TNF-α-treated astrocytes, respectively (Figure 5J).

### 2.4. The IL-1β Plus TNF-α-Induced Uptake of 2-NBDG Glucose via Cx43 Hemichannels Is Cooperative, Competitive and Saturable in Cultured Astrocytes

Previous studies have suggested that during pro-inflammatory conditions the influx of glucose could rely on the opening of Cx43 hemichannels in cultured astrocytes [42]. However, the permeability features of this process remain poorly understood. In light of this, we then examined several features of Cx43 hemichannel-dependent uptake of 2-NBDG, a fluorescent glucose analogue. Control astrocytes bathed with normal concentrations of Ca^2+^/Mg^2+^ showed a basal uptake of 2-NBDG that was drastically blunted by 100 mM ETDG or 100 µM Cyto-B (Figure 6A–F), two GLUT1 inhibitors [45]. In contrast, both gap19 or Tat-L2 failed in eliciting a similar inhibitory response (Figure 6F), indicating that basal 2-NBDG uptake relies on GLUT1 in control astrocytes. To figure out the roles of Cx43 hemichannels we performed experiments in astrocytes bathed in the absence of extracellular Ca^2+^/Mg^2+^. We found that DCFS triggered a ~3-fold augment in 2-NBDG uptake compared to control that was completely suppressed by gap19 or Tat-L2 (Figure 6F). The relevance of Cx43 hemichannels in this response was confirmed by the fact that when exposed to DCFS, astrocytes from Cx43^−/−^ mice showed a similar 2-NBDG uptake than wild type astrocytes bathed with saline solution containing normal concentrations of Ca^2+^/Mg^2+^ (Figure 6F). Conversely, ETDG or Cyto-B partially blunted the DCFS-induced 2-NBDG uptake (Figure 6F), which likely corresponded to the basal 2-NBDG uptake via GLUT1 rather than Cx43 hemichannels.

IL-1β/TNF-α-treated astrocytes exhibited a ~4-fold increase in basal 2-NBDG uptake when compared to control conditions (Figure 6A–G). This response was fully tackled by gap19 or Tat-L2 and did not occur in astrocytes from Cx43^−/−^ mice, revealing the contribution of Cx43 hemichannels in that process (Figure 6G). Relevantly, DCFS caused a 2-fold increase in 2-NBDG uptake in IL-1β/TNF-α-treated astrocytes when compared to control astrocytes bathed with DCFS, this effect being totally reduced by gap19 or Tat-L2, but weakly inhibited by ETDG or Cyto-B (Figure 6G). Noteworthy, the DCFS-induced 2-NBDG uptake was not detected in Cx43^−/−^ astrocytes treated with Cyto-B, regardless the presence or absence of the treatment with IL-1β/TNF-α (Figure 6G). We further investigated whether 2-NBDG uptake via Cx43 hemichannels in astrocytes could exhibit cooperative features. As observed for Etd, the treatment with IL-1β/TNF-α left-shifted the sigmoidal relationship between the uptake of 2-NBDG and its concentration in astrocytes bathed with DCFS (Figure 6H). Consistent with this, IL-1β/TNF-α reduced from ~300 µM to ~200 µM the concentration to what the maximal 2-NBDG uptake rate reached (Figure 6H). In the same line, although IL-1β/TNF-α did not modify the V_max_ and Hill coefficient for 2-NBDG uptake, it reduced the K_d_ from 166.4 ± 24.7 to 73.4 ± 10.9 µM (Figure 6H). As seen for nucleic acid dyes, the uptake of 2-NBDG via Cx43 hemichannels also behave in a competitive manner. We noted that 150 µM Etd right-shifted the sigmoidal curve of 2-NBDG uptake without changing the V_max_ or Hill coefficient in control astrocytes, but increasing the K_d_ 166.4 ± 24.7 to 245.2 ± 36.7 µM (Figure 6H). Similar competitive tendencies evoked by Etd were found in astrocytes stimulated with IL-1β/TNF-α (Figure 6H). Indeed, in both control and IL-1β/TNF-α-treated astrocytes Etd elicited a concentration-dependent inhibition of 2-NBDG uptake rates with an IC_50_ of 44.4 ± 8.9 and 90.1 ± 13.8 µM, respectively (Figure 6I).

### 2.5. IL-1β Plus TNF-α Modulates the Gap Junctional-Mediated Intercellular Spread of Molecules Depending on the Permeant Species in Cultured Astrocytes

Gap junction-mediated intercellular communication among astrocytes is crucial for proper spatial buffering of extracellular K^+^, as well as the spread of energy substrates and intercellular Ca^2+^ waves at the nervous system [45]. There is mounting evidence indicating that pro-inflammatory conditions increase the activity of Cx43 hemichannels but reduce astrocyte–astrocyte cell coupling [21,24,42]. However, it is known that the charge selectivity of homotypic gap junction channels to different solutes is connexin-specific [46,47], whereas physical properties of permeants such as molecular size, charge and shape affect their passage through these channels [48,49,50,51]. Taking this into account, we explored whether IL-1β/TNF-α could affect the diffusion of different molecules through gap junctions in astrocytes (Figure 7).

Dye coupling experiments showed that IL-1β/TNF-α did not alter the gap junction-mediated intercellular diffusion of Etd (Figure 7A,B,K,L) or DAPI (Figure 7E,F,O,P), but it reduced the number of astrocytes coupled to Prd (Figure 7C,D,M,N) or 2-NBDG (Figure 7I,J,S,T) from ~7 to ~4 cells or ~17 to ~7 cells, respectively (Figure 7U). Of note, the same stimulus greatly increased astroglial coupling to YO-PRO-1 from ~5 to ~14 cells (Figure 7G,H,Q,R). These findings indicate that IL-1β/TNF-α could alter the permeation of molecules via astroglial Cx43 gap junction channels depending on the permeant species.

## 3. Discussion

In the last decade, mounting evidence has described that hemichannels serve as conduits for the exchange between both the cytoplasm and the extracellular milieu. Although most of these studies argue for a major role of these channels during pathological conditions [52], recent findings indicate that hemichannels also open in diverse biological processes within the nervous system [14,15,53]. What still remains poorly understood is how the flux of permeants through hemichannels is affected during physiological and pathological conditions. In this study, we provide the first evidence supporting that an inflammatory condition alters the uptake of cationic molecules via astroglial Cx43 hemichannels depending on the properties of the permeant species.

Time-lapse recordings performed without extracellular Ca^2+^ and Mg^2+^—a stimulus that increases the open probability of connexin hemichannels—revealed a positive cooperative uptake for low concentrations of Etd, Prd, DAPI or YO-PRO-1 and saturation at higher concentrations. The DCFS-induced uptake for all dyes studied was greatly reduced by two specific Cx43 hemichannel mimetic peptides (gap19 and Tat-L2), but not by their inactive forms (gap19^I130A^ or Tat-L2^H126K/I130N^). The crucial role of astroglial Cx43 hemichannels as conduits for small cationic molecules was corroborated by the undetectable significant changes in dye uptake in Cx43^−/−^ astrocytes exposed to DCFS. The latter harmonizes with the strong correlation between Etd, Prd, DAPI or YO-PRO-1 uptake rates and Cx43 expression amounts, either in the presence or absence of extracellular Ca^2+^ and Mg^2+^.

The relationship between dye uptake rate and dye concentration for all tracers resulted in sigmoidal (or “S-shaped”) curves with Hill coefficients >1, suggesting positive cooperative influx of the same permeant, and revealing that astroglial Cx43 hemichannels might contain distinct binding sites for Etd, Prd, DAPI or YO-PRO, respectively. Most likely, the V_max_ observed for all dyes studied was not due to saturation of intracellular binding sites of nucleic acids, as permeabilization of the cell membrane with 70% ethanol triggered a striking rise in dye uptake rate to levels far above the peak value observed with unpermeabilized cells. In addition, we found that DAPI and Etd cross the astroglial membrane via Cx43 hemichannels, as their simultaneous presence reduced the K_d_ of each other without altering their respective V_max_, unveiling a competitive process. This evidence is consistent with the fact that Cx43-EGFP hemichannels expressed in HeLa cell show cooperativity, saturation and competitive features for the uptake of Etd, Prd and DAPI [26]. Although the above coherence could seem trivial, other groups have failed in finding Cx43 hemichannel-dependent Etd uptake in astrocytes or Cx43-expressing C6 cells, regardless of observing the opposite in Cx43-expressing *Xenopus* oocytes [27,29]. The discrepancy between our findings and those studies concerning Etd uptake permeability may rely on differences in culture conditions, including cell confluency, time of culture, inflammatory profile, presence of gliotransmitters and other paracrine factors and redox and the metabolic state.

Despite pro-inflammatory agents augmenting the activity of Cx43 hemichannels in astrocytes [23,25,42,54], studies addressing if the kinetic of permeation to molecules is affected during these conditions are lacking. Here, we showed that IL-1β/TNF-α, well-known mix of proinflammatory cytokines that increase the activity of astroglial Cx43 hemichannels, shift the S-shaped curves of dye uptake depending on the permeant molecules. In particular, IL-1β/TNF-α left-shifted the dye uptake rate vs. dye concentration relationship for Etd, but the opposite took place for DAPI or YO-PRO-1, whereas no alterations were observed for Prd. The latter was accompanied of changes in K_d_ (Etd, DAPI or YO-PRO-1) and Hill coefficients (Etd and YO-PRO-1), without significant changes in V_max_. This evidence suggests that IL-1β/TNF-α may distinctively affect the binding sites to permeants in astroglial Cx43 hemichannels rather than their number at the cell surface. Alternatively, IL-1β/TNF-α could induce the production of biological permeants that may favor or compete for in the pore-lining residues of Cx43 hemichannels. A putative mechanism could involve changes in posttranscriptional modification, including changes in redox and/or phosphorylation state of Cx43 [55].

Our study brings to light the current debate about the long-time belief that hemichannels and other large-pore channels act as free diffusion routes for ions and molecules [56]. Recent antecedents have shown that permeability to small molecules in large-pore channels, including hemichannels, does not always match with the atomic ion permeation and conversely [27,28,29,57]. With this in mind, it is possible to speculate that affinity and/or binding sites towards ions and small molecules may be different within the pore-lining region of hemichannels. This could explain why under physiological circumstances Cx43 hemichannels allow the release of gliotransmitters without allowing the influx of extracellular Ca^2+^ [53], whereas in pathological conditions, they could be permeable to both [25,58]. Thus, in the normal brain, if the uptake of molecules and/or ions shares equivalent kinetic properties as that of release, competitive inhibition might mitigate the loss of important amounts of critical molecules for cell survival. On the other hand, during pathological scenarios, alterations on permeability properties of Cx43 hemichannels may affect the normal balance of influx and release of molecules and/or ions with potentially significant repercussions for the pathogenesis of diseases.

How the kinetics of hemichannel permeability properties are related to the permeation of molecules through gap junctions is an issue that remains poorly studied. Using Lucifer yellow (−2) and Etd (+1) as permeability probes to assess the functional state of gap junctions and hemichannels, respectively, an opposite response has been observed upon the action of inflammatory conditions [42,59]. Notably, in those studies, the activities of Cx43 hemichannels and gap junction channels were assayed with different permeability probes and since these studies were complemented with results from different experimental approaches (e.g., electrophysiology, protein membrane biotinylation), their interpretations are likely to remain valid. In fact, if the activity of Cx43 hemichannels would have been evaluated with Prd, DAPI or YO-PRO-1 the outcome would have been a reduction in Cx43 hemichannel activity, and in the same direction of the gap junction changes detected with Lucifer yellow. Similarly, if the functional state of gap junctions would have been evaluated with Etd, DAPI or Prd, no significant changes would have been detected and conversely, the use of YO-PRO-1 would have revealed an increase in dye coupling and only results with 2-NBDG would have been similar to those obtained with Lucifer yellow. This argument suggests that studies on permeability changes of connexin-based channels obtained only with one molecular probe per channel type should be taken cautiously and certainly, the use of diverse experimental approaches could ascertain a better understanding of the problem under study.

Although glucose is mostly transported to the cytoplasm via GLUTs in astrocytes [60], the conditioned media of LPS-treated microglia shift part of this influx through Cx43 hemichannels [42]. Here, we observed that 2-NBDG crosses the plasma membrane of astrocytes stimulated with these cytokines, which are likely to mediate the increase in 2-NBDG uptake of astrocytes cultured for 24 h with conditioned medium by LPS-activated microglia [42]. Herein, we also showed that the 2-NBDG uptake via Cx43 hemichannels is cooperative and competitive. The increase in 2-NBDG uptake occurred without changes in Hill coefficient and V_max_ but with reduction in K_d_, suggesting and increase affinity for 2-NBGD, which might explain the increase in 2-NBGD uptake promoted by TNF-α/IL-1β. The kinetic parameters of 2-NBDG permeation might be extrapolated to molecules of biological relevance. From this perspective, the existence of molecules that compete with glucose for binding sites at the Cx43 hemichannel pore will favor the transport of this metabolite through GLUTs, as we observed in astrocytes bathed with normal saline. In contrast, the absence of competitive molecules, which could occur upon treatment with IL-1β/TNF-α, can lead to a significant uptake of glucose through Cx43 hemichannels. The latter line of thought is supported by the competitive effect that Etd elicited on 2-NBDG uptake in both control and IL-1β/TNF-treated astrocytes.

As mentioned previously, the intercellular passage of small molecules via gap junctions is regulated by IL-1β/TNF-α [42,61] and here, we demonstrated that involves a mechanism that is permeant-dependent. A possible limitation of dye coupling measurements could derive from features of intracellular binding sites and their possible modifications upon particular treatment. Changes of that kind could limit in different degree the simple diffusion of dyes from one cell to another because the driving force (concentration gradient of the permeability tracer) could vary in different degrees if the affinities for those binding sites are affected. Whether gap junctions exhibit similar cooperative and competitive kinetics, as those showed here for Cx43 hemichannels, remains unknown, but it is highly likely that is the case. Certainly, the presence of those properties for molecular permeation would impact complex tasks at the synaptic cleft, including the delivery of lactate and glucose, NAD^+^ paracrine signaling, buffering of extracellular K^+^ and glutamate, as well as spread of intercellular Ca^2+^ waves [4,5,6].

The idea that certain stimuli may induce opposite changes in dye uptake kinetics of hemichannels depending on the molecule studied, along with the mismatch between ionic vs. small molecule permeation, raise the need of pursuing new methodological approaches to study these channels. Future studies will elucidate whether the differential ionic/molecule permeation of Cx43 hemichannels in astrocytes could impact their communication with neurons in the normal and inflamed nervous system.

## 4. Materials and Methods

### 4.1. Reagents and Antibodies

HEPES, water (W3500), Dulbecco′s Modified Eagle Medium (DMEM), 4,6,-O-ethylidene-D-glucose (ETDG), anti-GFAP polyclonal antibody, cytosine arabinoside (Ara-C), LaCl_3_ (La^3+^), cytochalasin B (Cyto-B) and ethidium (Etd) bromide were purchased from Sigma-Aldrich (St. Louis, MO, USA). Fetal bovine serum (FBS) was purchased from Hyclone (Logan, UT, USA), whereas penicillin, 2-(N-(7-nitrobenz-2-oxa-1,3-diazol-4-yl)amino)-2-deoxyglucose (2-NBDG), streptomycin, Propidium (Prd) iodide, diamidino-2-phenylindole (DAPI), YO-PRO-1, goat anti-mouse Alexa Fluor 488/555 and goat anti-rabbit Alexa Fluor 488/555 were from Thermo Fisher Scientific (Waltham, MA, USA). IL-1β and TNF-α were obtained from Roche Diagnostics (Indianapolis, MI, USA). Normal goat serum (NGS) was purchased from Zymed (San Francisco, CA, USA). Anti-Cx43 monoclonal antibody (610061) was obtained from BD Biosciences (Franklin Lakes, NJ, USA). The mimetic peptides gap19 (KQIEIKKFK, intracellular loop domain of Cx43), gap19^I130A^ (KQAEIKKFK, negative control), Tat-L2 (YGRKKRRQRRR-DGANVDMHLKQIEIKKFKYGIEEHGK, intracellular loop domain of Cx43), and Tat-L2^H126K/I130N^ (YGRKKRRQRRR-DGANVDMKLKQNEIKKFKYGIEEHGK, negative control) were obtained from Genscript (Piscataway, NJ, USA).

### 4.2. Cortical Astrocytes

Primary astrocytes were prepared from the cortex of postnatal day 2 C57BL/6 mice as previously reported [24]. Animal protocols were conducted following the guideline and approved protocol for care and use of experimental animals of the Bioethics Committee of the Pontificia Universidad Católica de Chile (PUC, No: 150806013, 6 June 2016) and the European Community Council Directives of November 24th, 1986. In short, upon brain removal and dissection of cortices, meninges were carefully peeled off and tissue was mechanically dissociated in Ca^2+^- and Mg^2+^-free Hank′s balanced salt solution with 0.25% trypsin and 1% DNase. Cells were seeded onto 60 mm plastic dishes (Corning, NY, USA) or onto glass coverslips (Fisher Scientific, Waltham, MA, USA) placed inside 16-mm 24-well plastic plates (Corning, NY, USA) at a density of 2 × 10^6^ cells/dish or 1 × 10^5^ cells/well, respectively, in DMEM, supplemented with penicillin (5 U/mL), streptomycin (5 µg/mL), and 10% FBS. Cells were grown at 37 °C in a 5% CO_2_/95% air atmosphere at nearly 100% relative humidity. Following 8–10 days in vitro (DIV), 1 µM AraC was added for 3 days in order to suppress the proliferation of microglia. Medium was changed twice a week and cultures were used after 3 weeks. In some experiments, cortical primary Cx43-deficient astrocytes were obtained from male Cx43 knock-out (KO) (Cx43^−/−^) mice, whereas control Cx43^+/+^ wild-type astrocytes, were cultured from male mice with the same genetic background [62]. Cx43^−/−^ and Cx43^+/+^ mutant mice were generated by mating between heterozygous Cx43^+/−^.

### 4.3. Treatments

Astrocytes were treated for 24 h with a mixture of IL-1β and TNF-α (10 ng/mL for each). Mimetic peptides against Cx43 hemichannels (gap19, gap19^I130A^, Tat-L2 or Tat-L2^H126K/I130N^, 100 µM) were applied to cell cultures 15 min prior to and co-applied during dye uptake and time-lapse recordings (see below). Similarly, in another set of experiments, two glucose transporter (GLUT) inhibitors, 100 µM Cyto-B or 100 mM ETDG [63] were applied to cell cultures 15 min prior to and co-applied during 2-NBDG uptake experiments.

### 4.4. Dye Uptake and Time-Lapse Fluorescence Imaging

For time-lapse fluorescence imaging, astrocytes plated on glass coverslips were washed twice in Hank′s balanced salt solution and bathed at room temperature with recording solution (in mM: 148 NaCl, 5 KCl, 1.8 CaCl_2_, 1 MgCl_2_, 5 glucose and 5 HEPES, pH 7.4) in the presence of 5 µM Etd, Prd, DAPI or YO-PRO-1. Then, astrocytes were mounted on the stage of an Olympus BX 51W1I upright microscope with a 40x water immersion objective for time-lapse imaging. In some experiments, uptake of Etd, Prd, DAPI or YO-PRO-1 was measured in astrocytes exposed to a Ca^2+^- and Mg^2+^- free recording solution supplemented with 10 mM EGTA (DCFS, divalent cation-free solution). The latter is a well-accepted condition that increases the open probability of Cx43 hemichannels [37]. Images were captured by a Retiga 1300I fast-cooled monochromatic digital camera (12-bit) (Qimaging, Burnaby, BC, Canada) controlled by imaging Metafluor software (Universal Imaging, Downingtown, PA) every 30 s (exposure time = 0.5 s; excitation and emission wavelengths were 528 and 598 nm, respectively for the case Etd and Prd fluorescence, whereas for DAPI or YO-PRO-1 fluorescence the excitation and emission wavelengths were 355 and 405 nm or 480 and 520 nm, respectively). The fluorescence intensity recorded from ≥30 regions of interest (representing at least 30 cells per cultured coverslip) was calculated with the following formula: Corrected total cell fluorescence = Integrated Density − ([Area of the selected cell] × [Mean fluorescence of background readings]). The dye uptake rate represents the mean slope of the relationship over a given time interval (ΔF/ΔT). To explore the changes in the slope, regression lines were fitted to points before and after the different experimental conditions using the Excel program, and mean slope values were analyzed using GraphPad Prism software and expressed as AU/min. Binding constants for each dye were calculated using the following equation for allosteric sigmoidal binding kinetics: Y = (V_max_ [S]^h^)/(K_d_ + [S]^h^), where V_max_ represents the maximal transport velocity, [S] the concentration of permeant, h the Hill coefficient, and K_d_ the permeant concentration needed to achieve half-maximal transport velocity. In concentration-response inhibition experiments, the IC50 was determined with the [inhibitor] vs. normalized response-variable slope regression according to the following equation: Y = 100/(1 + ([S]^HillSlope^)/(IC_50_^HillSlope^)), where [S] the concentration of permeant, Hill slope is the steepness of the curve, which has no unit and IC_50_ is the concentration of permeant that gives a response half way between bottom and top.

### 4.5. Dye Coupling

Astrocytes plated on glass coverslips were bathed with a recording medium (HCO_3-_-free F-12 medium buffered with 10 mM HEPES, pH 7.2) and permeability mediated by gap junctions was tested by evaluating the transfer of Etd, Prd, DAPI or YO-PRO-1 to neighboring cells. Briefly, astrocytes were iontophoretically microinjected with a glass micropipette filled with 10 mM Etd, Prd, DAPI or YO-PRO-1 in recording medium containing 200 μM La^3+^. This connexin hemichannel blocker employed to prevent cell leakage of the microinjected dye via hemichannels, which could underestimate dye transfer to neighboring cells. Fluorescent cells were observed using a Nikon inverted microscope equipped with epifluorescence illumination (Xenon arc lamp) and a Nikon B filter for dyes (excitation wavelength 528 nm; emission wavelength above 598 nm for Etd and Prd, whereas for DAPI or YO-PRO-1 fluorescence the excitation and emission wavelengths were 355 and 405 nm or 480 and 520 nm, respectively) and an XF34 filter to DiI fluorescence (Omega Optical, Inc., Brattleboro, VT, USA). Photomicrographs were obtained using a CCD monochrome camera (CFW-1310M; Scion; Frederick, MD, USA). Three minutes after dye injection, cells were observed to determine whether dye transfer occurred. The coupling index was scored as the mean number of cells to which the dye spread from the injected cell to more than one neighboring cell. Three experiments were performed for every treatment and dye coupling was tested by microinjecting a minimum of 10 cells per experiment.

### 4.6. Immunofluorescence and Confocal Microscopy

Astrocytes grown on coverslips were fixed at room temperature with 2% paraformaldehyde (PFA) for 30 min and then washed three times with PBS. They were incubated three times for 5 min in 0.1 M PBS-glycine, and then, in 0.1% Triton X-100 in PBS containing 10% NGS for 30 min. Cells were incubated with an anti-GFAP polyclonal antibody (Sigma-Aldrich, St. Louis, MO, USA, 1:400) or anti-Cx43 monoclonal antibody (BD Biosciences, Franklin Lakes, NJ, USA, 1:400) diluted in 0.1% Triton X-100 in PBS with 2% NGS at 4 °C overnight. After five rinses in 0.1% Triton X-100 in PBS, cells were incubated with goat anti-mouse IgG Alexa Fluor 355 (1:1000) or goat anti-rabbit IgG Alexa Fluor 488 (1:1000) at room temperature for 50 min. After several rinses, coverslips were mounted in DAKO fluorescent mounting medium and examined with an Olympus BX 51W1I upright microscope with a 40X water immersion or a confocal system Nikon Eclipse C2 up microscope with 60X. Nuclei were stained with DAPI. To assess the fluorescent intensity of Cx43 at the cell margin, stacks of consecutive confocal images were taken with a confocal system Nikon Eclipse C2 up microscope and a 60X oil immersion objective (1.4 NA) at 200 nm intervals. Images were acquired sequentially with three lasers (in nm: 408, 488 and 543), and Z projections were reconstructed using Nikon confocal software NIS-elements (Nikon Instruments Inc, Melville, NY, USA). Image analysis of Z projections was then performed with ImageJ software. Cx43 signal intensity in both cell margin and cytoplasm was calculated with the following formula: Corrected fluorescence = Integrated Density − ([Area of selected cell] × [Mean fluorescence of background readings]).

### 4.7. Uptake and Intercellular Diffusion of 2-NBDG Glucose

Astrocytes plated on glass coverslips were incubated with 100 μM 2-NBDG, a glucose analogue (342.3 g/mol, neutral, Thermo Fisher, Carlsbad, CA, USA), for 30 min at room temperature in Locke’s solution (in mM: 154 NaCl, 5.4 KCl, 2.3 CaCl_2_, 5 HEPES, pH 7.4). Then, astrocytes were mounted on the stage of a confocal Leica SP5 MP microscope with a 40x water immersion objective for time-lapse imaging controlled by the Leica Acquisition software. In some experiments, 2-NBDG uptake was measured in astrocytes exposed to DCFS. Emitted fluorescence was detected using two non-descanned detectors, with a 525/50 nm filter and captured every 30 s. The fluorescence intensity recorded from ≥30 regions of interest (representing at least 30 cells per cultured coverslip) was calculated with the following formula: Corrected total cell fluorescence = Integrated Density − ([Area of the selected cell] × [Mean fluorescence of background readings]). The dye uptake rate represents the mean slope of the relationship over a given time interval (ΔF/ΔT). To explore the changes in the slope, regression lines were fitted to points before and after the different experimental conditions using Excel program, and mean slope values were analyzed using GraphPad Prism software and expressed as AU/min. Affinity constants for each dye were calculated using the following equation for allosteric sigmoidal binding kinetics: Y = (V_max_ [S]^h^)/(K_d_ + [S]^h^), where V_max_ represents the maximal transport velocity, [S] the concentration of permeant, h the Hill coefficient, and K_d_ the permeant concentration needed to achieve half-maximal transport velocity. Each independent experiment embraced three replicates. In some experiments, cultured astrocytes were pre-incubated with synthetic mimetic peptides gap19 (100 µM) or Tat-L2 (100 µM), as well as with the GLUT1 inhibitors ETDG (100 mM) or Cyto-B (100 µM) for 15 min before and during the dye uptake experiments.

In order to evaluate the intercellular diffusion of 2-NBDG via gap junctions to neighboring cells, astrocytes were loaded with 2-NBDG by iontophoresis in the following extracellular solution: in mM 126 NaCl, 2.5 KCl, 1.25 NaH_2_PO_4_, 26 NaHCO_3_, 20 glucose, 5 pyruvate, 2 CaCl_2_ and 1 MgCl_2_ (95% O_2_, 5% CO_2_). Briefly, astrocyte visualized by DIC under an epifluorescence microscope (Leica DM FLS, Leica microsystems, Germany) were impaled with a patch-clamp glass micropipette (~1 μm tip diameter, typical resistance of ~7 MΩ) filled with 20 mM 2-NBDG. Under whole cell configuration (Axopatch 200B, Axon instruments, San Jose, CA, USA) short voltage step pulses were applied to favor the dialysis of the dye into the injected astrocyte. The extracellular medium was supplemented with La^3+^ (200 μM) to avoid cell leakage of the dye through connexin hemichannels. Fluorescent cells were observed by epifluorescence excitation at λ~480 nm being acquired at λ = 520 nm by using a video camera (CMOS Digital Camera 5MP, Micrometrics, Norcross, GA, USA). The coupling index was calculated as the mean number of cells to which the 2-NBDG spread. Three experiments were performed for every treatment and 2-NBDG coupling was tested by microinjecting a minimum of 10 cells per experiment.

### 4.8. Data Analysis and Statistics

For each data group, results were expressed as mean ± standard error (SEM); n refers to the number of independent experiments. Detailed statistical results were included in the figure legends. Statistical analyses were performed using GraphPad Prism (version 7, GraphPad Software, La Jolla, CA, USA). Normality and equal variances were assessed by the Shapiro–Wilk normality test and Brown–Forsythe test, respectively. Unless otherwise stated, data that passed these tests were analyzed by an unpaired t-test in the case of comparing two groups, whereas in the case of multiple comparisons, data were analyzed by one or two-way analysis of variance (ANOVA) followed, in the case of significance, by Tukey’s post hoc test. When data were heteroscedastic as well as not normal and with unequal variances, we used the Mann–Whitney test in case of comparing two groups, whereas in the case of multiple comparisons, data were analyzed by Kruskal–Wallis test followed, in the case of significance, by Dunn′s post hoc test. A probability of *p* < 0.05 was considered statistically significant.

## Figures and Tables

**Figure 1 ijms-21-03970-f001:**
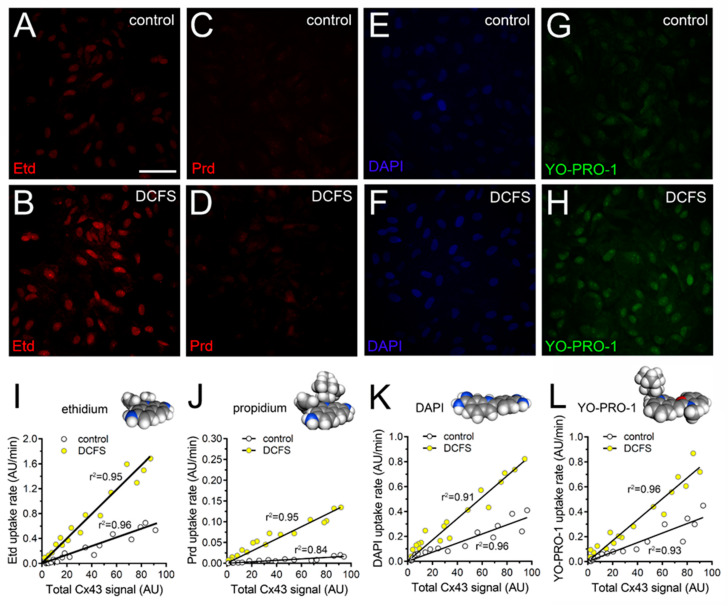
Uptake of nucleic acid dyes is directly proportional to total amount of Cx43 in cultured astrocytes. (**A**–**H**) Representative fluorescence images depicting Etd (**A**,**B**), Prd (**C**,**D**), DAPI (**E**,**F**), and YO-PRO-1 (**G**,**H**) staining from dye uptake measurements (10 min exposure to each dye) in astrocytes bathed in control saline or a divalent cation-free solution (DCFS). (**I**–**L**) Representative plots of linear regression analysis showing the close relationship between uptake rate of Etd (**I**), Prd (**J**), DAPI (**K**) or YO-PRO-1 (**L**) and total immunostaining signal for Cx43 in astrocytes bathed in control saline (white circles) or DCFS (yellow circles). The values for Pearson′s r were statistically significant (*p* < 0.0001) for all dyes studied: Etd (control: 0.96; DCFS: 0.95), Prd (control: 0.84; DCFS: 0.96), DAPI (control: 0.93; DCFS: 0.96) or YO-PRO-1 (control: 0.94; DCFS: 0.96). Space-filling models for the atomic structure of each dye were elaborated with the MolView GPL software (carbon: gray, hydrogen: dark gray, and nitrogen: blue). Calibration bar: 50 µm.

**Figure 2 ijms-21-03970-f002:**
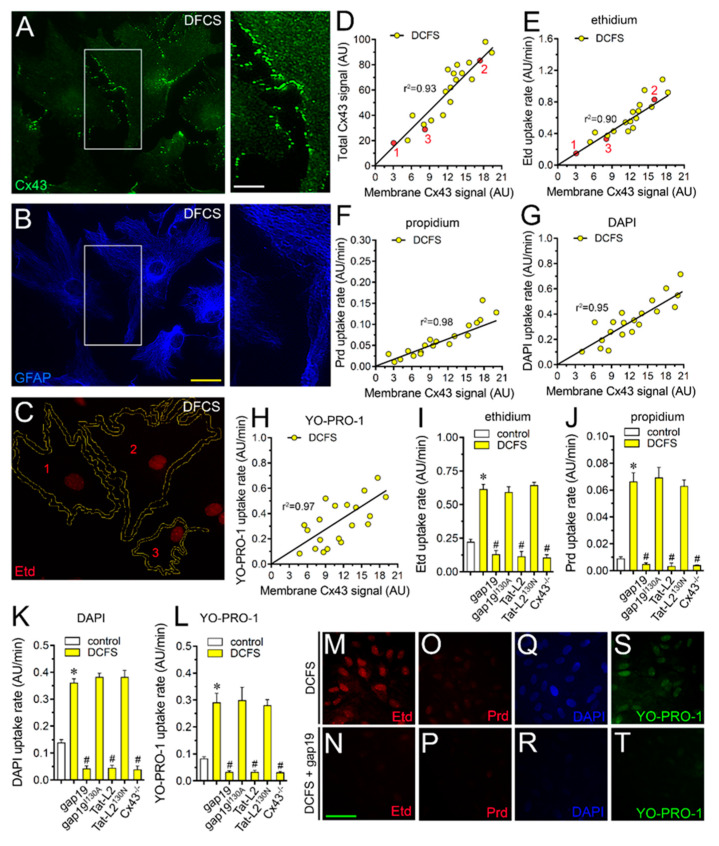
Uptake of nucleic acid dyes depends on Cx43 hemichannels in cultured astrocytes exposed to a divalent cation-free solution. (**A**–**C**) Representative confocal images depicting Cx43 (green, **A**), GFAP (blue, **B**) and Etd (red, **C**) staining by astrocytes bathed with DCFS for 10 min in the presence of 5 µM Etd. Right insets: 1.8× magnification of the indicated area of panels A and B. (**D**) Representative linear regression analysis showing the relationship between the total and cell periphery (cell surrounding) immunostaining signal of Cx43 in cultured astrocytes bathed with DCFS (yellow circles). Note that red points in the graph correspond to the Cx43 fluorescence values between inner and outer yellow lines shown for cells 1–3 in panel C. Pearson′s r was 0.94 (*p* < 0.0001). (**E**–**H**) Representative plots of linear regression analysis showing the relationship between uptake rate of Etd (**E**), Prd (**F**), DAPI (**G**) or YO-PRO-1 (**H**) and intensity of membrane immunostaining signal for Cx43 in astrocytes bathed with DCFS (yellow circles). The values for Pearson’s r were statistically significant (*p* < 0.0001) for all dyes studied: Etd (0.95), Prd (0.96), DAPI (0.96) or YO-PRO-1 (0.96). (**I**–**L**) Averaged uptake rate of Etd (**I**), Prd (**J**), DAPI (**K**) or YO-PRO-1 (**L**) of astrocytes bathed with control saline (white bars) or DCFS (yellow bars). Cells were exposed for 5 min to 5 µM of each dye. It is also shown the response of astrocytes knock-out for Cx43 or the impact of the following pharmacological agents: 100 µM gap19, 100 µM gap19^I130A^, 100 µM Tat-L2 or 100 µM Tat-L2^H126K/I130N^. * *p* < 0.05, DCFS vs. control; ^#^
*p* < 0.05, pharmacological agents vs. DFCS (two-way ANOVA followed by Tukey’s post hoc test). (**M**–**S**) Representative fluorescence images depicting Etd (**M**,**N**), Prd (**O**,**P**), DAPI (**Q**,**R**), and YO-PRO-1 (**S**,**T**) staining from dye uptake measurements (10 min exposure to each dye) in astrocytes exposed to DCFS alone (upper panels) or plus 100 µM gap19 (below panels). Calibration bars: white 6 µm; yellow: 20 µm; and green: 40 µm.

**Figure 3 ijms-21-03970-f003:**
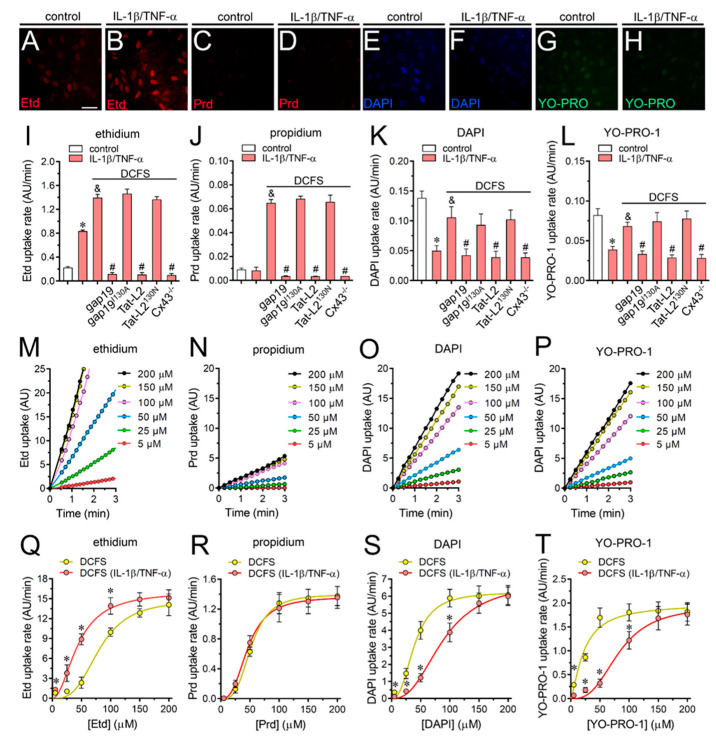
IL-1β plus TNF-α alters the cooperative permeation of dyes through Cx43 hemichannels depending on the permeant species in cultured astrocytes. (**A**–**H**) Representative fluorescence images depicting Etd (**A**,**B**), Prd (**C**,**D**), DAPI (**E**,**F**), and YO-PRO-1 (**G**,**H**) staining from dye uptake measurements (10 min exposure to each dye) in astrocytes under control conditions or after 24 h of treatment with IL-1β/TNF-α. (**I**–**L**) Averaged uptake rate of Etd (**I**), Prd (**J**), DAPI (**K**) or YO-PRO-1 (**L**) of control (white bars) or IL-1β/TNF-α-treated (red bars) astrocytes bathed with normal saline or DCFS. Cells were exposed for 5 min to 5 µM of each dye. It is also shown the response of astrocytes knock-out for Cx43 or the impact of the following pharmacological agents: 100 µM gap19, 100 µM gap19^I130A^, 100 µM Tat-L2 or 100 µM Tat-L2^H126K/I130N^. * *p* < 0.05, IL-1β/TNF-α 24 h vs. control; ^&^
*p* < 0.05, IL-1β/TNF-α vs. IL-1β/TNF-α plus DFCS; ^#^
*p* < 0.05, pharmacological agents vs. IL-1β/TNF-α plus DFCS (two-way ANOVA followed by Tukey’s post hoc test). (**M**–**P**) Representative time-lapse measurements of fluorescence intensity showing Etd (**M**), Prd (**N**), DAPI (**O**) or YO-PRO-1 (**P**) uptake of astrocytes in DCFS containing different increasing concentrations of the respective permeability tracer. (**Q**–**T**) Relationship between increasing concentrations of Etd (**Q**), Prd (**R**), DAPI (**S**) or YO-PRO-1 (**T**) and dye uptake rate of astrocytes bathed with DCFS under control conditions (yellow circles) or after 24 h treatment with IL-1β/TNF-α (red circles). Data were fitted using an empirical equation for fitting sigmoidal binding-velocity curves (see methods). * *p* < 0.05, control DCFS vs. IL-1β/TNF-α plus DFCS (two-way ANOVA followed by Tukey’s post hoc test). Data were obtained from at least three independent experiments with four repeats for each one (≥25 cells analyzed for each repeat). Calibration bar: 25 µm.

**Figure 4 ijms-21-03970-f004:**
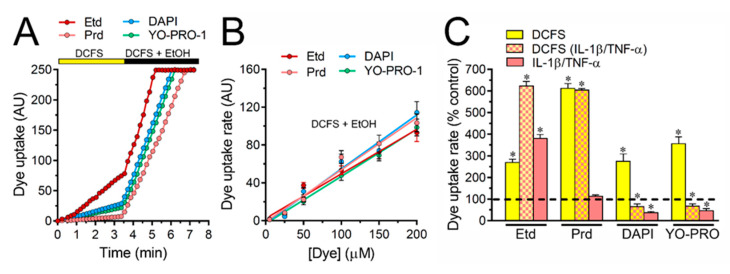
Maximum rate of dye uptake through astroglial Cx43 hemichannels is not due to saturation of intracellular binding sites. (**A**) Time-lapse measurements of Etd (dark red), Prd (light red), DAPI (blue) or YO-PRO-1 (green) uptake by astrocytes bathed in DCFS and then exposed to DCFS plus 70% *v*/*v* ethanol. Each value expresses the average of the fluorescence intensity in 20 cells; (**B**) Regression analysis showing the relationship between dye uptake rate and increasing concentrations of Etd (dark red), Prd (light red), DAPI (blue) or YO-PRO-1 (green) by astrocytes bathed with DCFS and permeabilized with 70% *v*/*v* ethanol. The fluorescence intensity was evaluated during initial velocity in three time periods separated by 15 s; (**C**) Dye uptake rate normalized to the amount of Cx43 reactivity found in the cell contour in control (yellow bars) or IL-1β/TNF-α-treated (red-yellow dashed bars) astrocytes bathed with DCFS. It is also shown the dye uptake by IL-1β/TNF-α-treated astrocytes bathed in normal saline (red bars). Cells were exposed for 5 min to 5 µM of each dye. * *p* < 0.05, respective control (dashed line) vs. treatments (two-way ANOVA followed by Tukey’s post hoc test). Data were obtained from at least three independent experiments with four repeats for each one (≥25 cells analyzed for each repeat).

**Figure 5 ijms-21-03970-f005:**
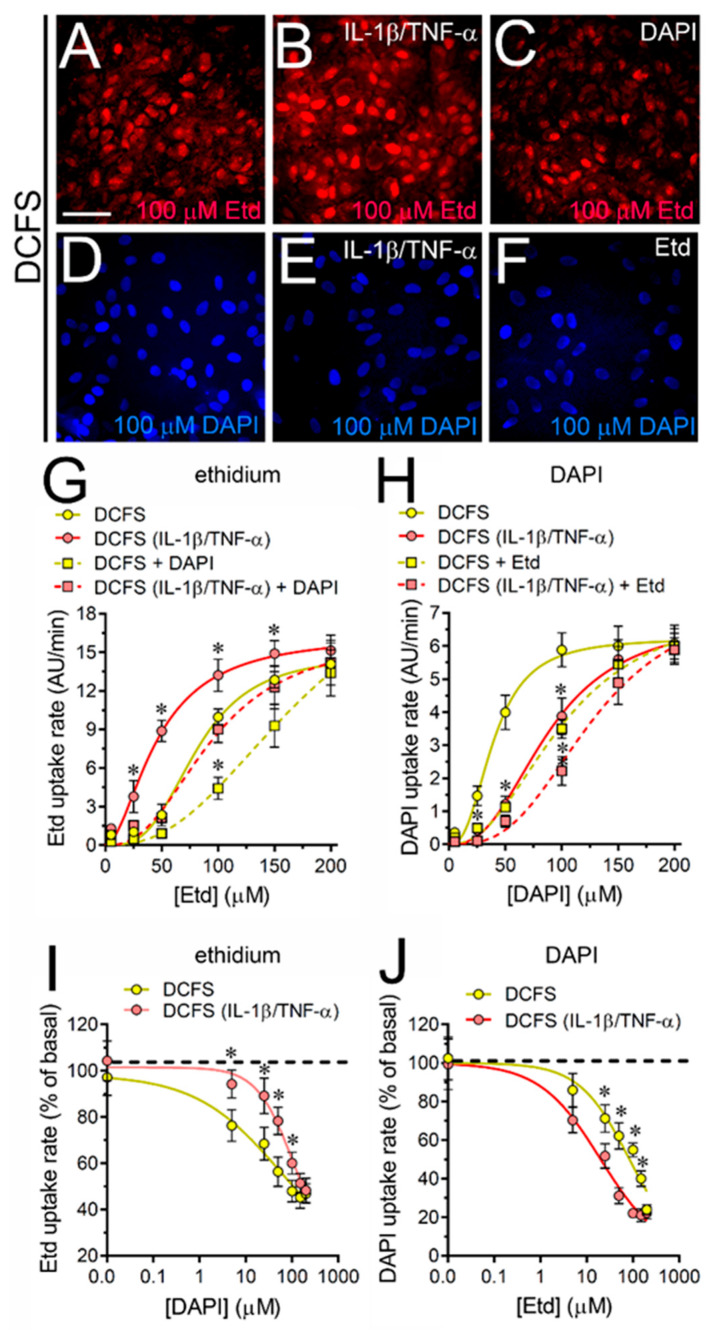
IL-1β plus TNF-α modulates the competitive permeation of cationic dyes through Cx43 hemichannels depending on the permeant species in cultured astrocytes. (**A**–**F**) Representative fluorescence images depicting Etd (**A**–**C**) and DAPI (**D**–**F**) staining from dye uptake measurements (10 min exposure to 100 µM each dye) in astrocytes bathed with DCFS under control conditions (**A**,**D**), after 24 h of treatment with IL-1β/TNF-α (**B**,**E**) or in the presence of 150 µM DAPI (**C**) or Etd (**F**). (**G**) Relationship between Etd uptake rate and increasing concentrations of Etd by astrocytes bathed with DCFS under control conditions alone (yellow circles) or plus 150 µM DAPI (yellow squares); or after treatment with IL-1β/TNF-α alone (red circles) or plus 150 µM DAPI (red squares). (**H**) Relationship between DAPI uptake rate and increasing concentrations of DAPI by astrocytes bathed with DCFS under control conditions alone (yellow circles) or plus 150 µM Etd (yellow squares); or after treatment with IL-1β/TNF-α alone (red circles) or plus 150 µM Etd (red squares). Data were fitted using an empirical equation for sigmoidal binding-velocity curves (see methods). * *p* < 0.05, control DCFS vs. treatments (two-way ANOVA followed by Tukey’s post hoc test). (**I**) Etd uptake rate normalized to the maximum effect evoked by DCFS in control (yellow circles) or IL-1β/TNF-α-treated (red circles) astrocytes exposed to increasing concentrations of DAPI. (**J**) DAPI uptake rate normalized to the maximum effect evoked by DCFS in control (yellow circles) or IL-1β/TNF-α-treated (red circles) astrocytes exposed to increasing concentrations of Etd. Data were fitted using the [inhibitor] vs. normalized response variable slope regression (see methods). * *p* < 0.05, control DCFS vs. IL-1β/TNF-α DCFS (two-way ANOVA followed by Tukey’s post hoc test). Data were obtained from at least three independent experiments with four repeats for each one (≥25 cells analyzed for each repeat). Calibration bar: 40 µm.

**Figure 6 ijms-21-03970-f006:**
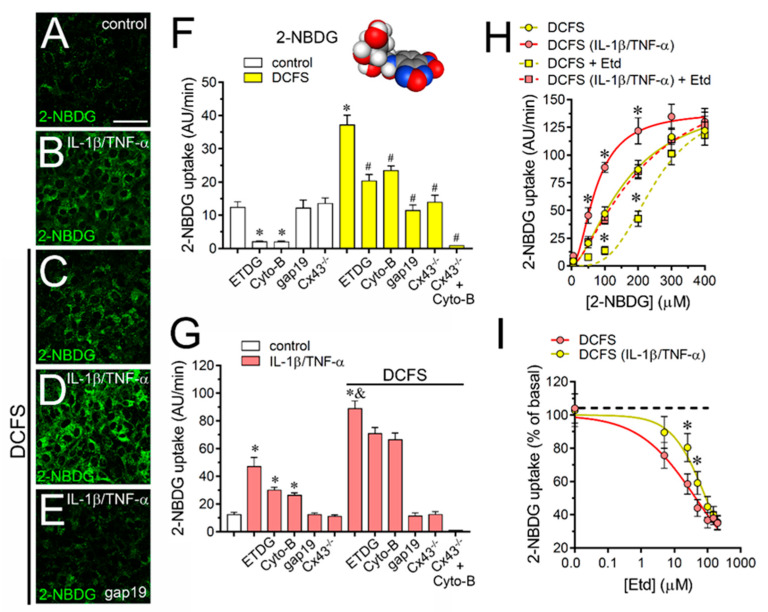
IL-1β plus TNF-α modifies permeation features of 2-NBDG glucose through Cx43 hemichannels in cultured astrocytes. (**A**–**E**) Representative fluorescence images depicting 2-NBDG staining from dye uptake measurements (10 min exposure to 100 µM 2-NBDG) in astrocytes under control conditions (**A**), after treatment with IL-1β/TNF-α (**B**), or bathed with DCFS under control conditions (**C**), or after 24 h of treatment with IL-1β/TNF-α alone (**D**) or plus 100 µM gap19 (**E**). (**F**) Averaged 2-NBDG uptake rate of astrocytes bathed with control saline (white bars) or DCFS (yellow bars). Cells were exposed for 10 min to 100 µM 2-NBDG. It is also shown the response of astrocytes knock-out for Cx43 or the impact of the following pharmacological agents: 100 mM ETDG, 100 µM Cyto-B, 100 µM gap19 or 100 µM Tat-L2. * *p* < 0.05, treatments vs. control; ^#^
*p* < 0.05, pharmacological agents vs. DCFS (two-way ANOVA followed by Tukey’s post hoc test). (**G**) Averaged 2-NBDG uptake rate by control (white bars) or IL-1β/TNF-α-treated (red bars) astrocytes bathed with normal saline or DCFS. Cells were exposed for 10 min to 100 µM 2-NBDG. It is also shown the response of astrocytes knock-out for Cx43 or the impact of the following pharmacological agents: 100 mM ETDG, 100 µM Cyto-B, 100 µM gap19 or 100 µM Tat-L2. * *p* < 0.05, IL-1β/TNF-α vs. control; ^&^
*p* < 0.05, IL-1β/TNF-α vs. IL-1β/TNF-α plus DCFS; ^#^
*p* < 0.05, pharmacological agents vs. IL-1β/TNF-α plus DCFS (two-way ANOVA followed by Tukey’s post hoc test). (**H**) Relationship between 2-NBDG uptake rate and increasing concentrations of 2-NBDG by astrocytes bathed with DCFS under control conditions alone (yellow circles) or plus 150 µM Etd (yellow squares); or after treatment with IL-1β/TNF-α alone (red circles) or plus 150 µM Etd (red squares). Data were fitted using an empirical equation for sigmoidal binding-velocity curves (see methods). * *p* < 0.05, control DCFS vs. treatments (two-way ANOVA followed by Tukey’s post hoc test). (**I**) 2-NBDG uptake rate normalized to the maximum effect evoked by DCFS in control (yellow circles) or IL-1β/TNF-α-treated (red circles) astrocytes exposed to increasing concentrations of Etd. Data were fitted using the [inhibitor] vs. normalized response variable slope regression (see methods). * *p* < 0.05, control DCFS vs. IL-1β/TNF-α DCFS (two-way ANOVA followed by Tukey’s post hoc test). Data were obtained from at least three independent experiments with four repeats for each one (≥25 cells analyzed for each repeat). Space-filling model for the atomic structure of 2-NBDG was elaborated with the MolView GPL software (carbon: gray, hydrogen: dark gray, and nitrogen: blue). Calibration bar: 20 µm.

**Figure 7 ijms-21-03970-f007:**
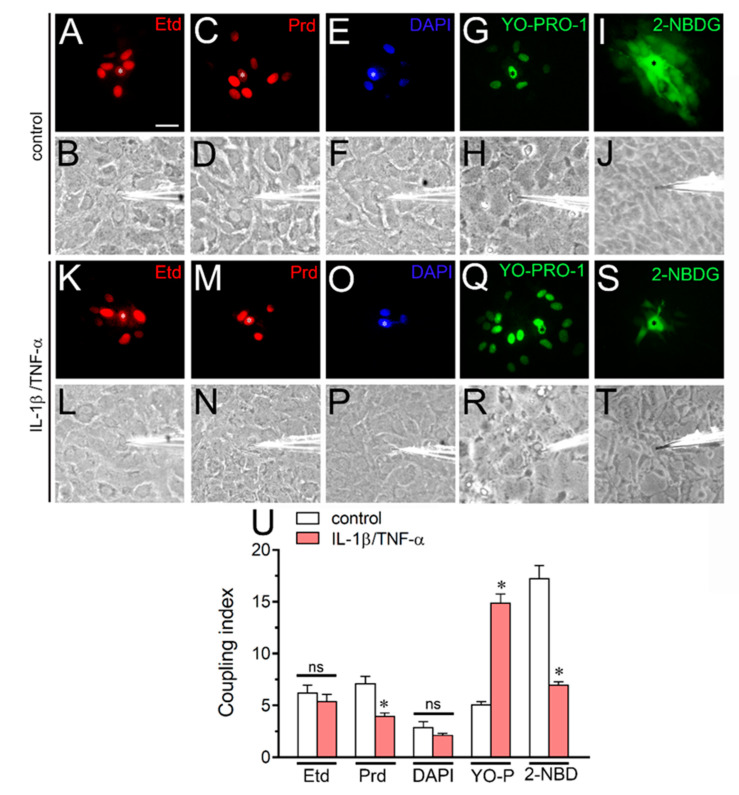
IL-1β plus TNF-α changes the intercellular diffusion of permeant molecules via Cx43 gap junction channels depending on the permeant species in cultured astrocytes. (**A**–**T**) Intercellular diffusion of Etd (**A**,**K**), Prd (**C**,**M**), DAPI (**E**,**O**), YO-PRO-1 (**G**,**Q**) or 2-NBDG (**I**,**S**) by astrocytes under control conditions or stimulated with IL-1β/TNF-α. In panels B, D, F, H, J, L, N, P, R and T are also shown the bright fields of each fluorescence picture. The * symbols indicate the injected cell. Calibration bar = 20 µm. (**U**) Coupling index (number of cells coupled in positive injections) for Etd, Prd, DAPI, YO-PRO-1 or 2-NBDG by astrocytes under control conditions (white bars) or stimulated with IL-1β/TNF-α (red bars). * *p* < 0.05, control vs. IL-1β/TNF-α (two-way ANOVA followed by Tukey’s post hoc test). Data were obtained from at least three independent experiments with three repeats for each one (≥10 cells analyzed for each repeat).

**Table 1 ijms-21-03970-t001:** Permeability parameters of astroglial Cx43 hemichannels.

	Ethidium		Propidium		DAPI		YO-PRO-1	
	Control	IL-1α/TNF-β	Control	IL-1α/TNF-β	Control	IL-1α/TNF-β	Control	IL-1α/TNF-β
V_max_ (AU/min)	14.8 ± 1.2	16 ± 1.3	1.5 ± 0.1	1.4 ± 0.1	6.2 ± 0.4	6.8 ± 0.7	2 ± 0.2	1.9 ± 0.2
Hill coefficient	3.2 ± 0.7	2.1 ± 0.6 *	3.1 ± 0.8	2.6 ± 0.9	2.6 ± 0.7	2.6 ± 0.5	1.4 ± 0.4	2.9 ± 0.8 *
K_d_ (µM)	80.9 ± 8	43.6 ± 6 *	54.3 ± 5.3	48.8 ± 7.2	39.5 ± 4.2	88.8 ± 11.1 *	22.5 ± 5.5	82.9 ± 11.1 *

* *p* < 0.05; IL-1β/TNF-α vs. control.

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
