# Peer review of "Permeation of Molecules through Astroglial Connexin 43 Hemichannels Is Modulated by Cytokines with Parameters Depending on the Permeant Species"

_ijms, 2020, doi:10.3390/ijms21113970_

Round 1

Reviewer 1 Report

The paper entitled “Permeation of molecules through astroglial connexin43 hemichannels is modulated by cytokines with parameters depending on the permeant species” aims to investigate whether the treatment with TNFalpha and IL1B alters astroglial Cx43 hemichannels permeability to different molecules.

Despite the introduction is really good focus and easy to understand, the results section needs a little bit of work on the authors side. I found that the description of the results is not really clear, and the data presentation is confusing most of the times. I suggest the authors to deep read the paper since I found pieces of text that have been moved from where they should be (e.g. from line 370-381).

Major comments

  1. Please show some picture regarding molecules uptake. If the experiments were performed by using time-lapse fluorescence imaging after dye-injection pictures should be shown. Please provide representative pictures for all the dye experiments (Fig 1, Fig 2, Fig 4 and Fig 5).

  1. When Cx43 IF was performed and analyzed by confocal microscopy (Fig 1F), what are the green dots that do not correspond to any cells? Can the author show a better image and lower magnification to fully convince that the green dots correspond to Cx43. 

  1. Figure 2. I suggest the authors to re-do this figure since the way it is presented it is really confusing according to what is written in the text. Could you please separate in different graphs the effect of gap19 and Tat-L2, the effect when added TNFalpha and IL1N, and the mice results? Also, revise figure mentioning in the text, I found no explanation for Fig 2 I-L. In this line, do the author consider that graphs shown in Fig 2 E-H are sigmoidal? Please, revise and correct in the text.

  1.  Please explain whythe authors treat the cells with 70% Ethanol, it seems no clear in the text.

  1. Figure 5. Same problem as in fig2, data presentation is confusing. Again, could you please separate in different graphs the results corresponding to the inhibitors and mice cells?

  1. From line 322 to 335, figure reference in the text should be in bold.

  1. Figure 6. What does * in the pictures means? Is refers to the injected cell? Please provide this information in the text or legend. How was the quantification done? By counting positive cells? I would like to see bright field images showing cell confluence.

Author Response

Reviewer 1.

*The paper entitled “Permeation of molecules through astroglial connexin43 hemichannels is modulated by cytokines with parameters depending on the permeant species” aims to investigate whether the treatment with TNFalpha and IL1B alters astroglial Cx43 hemichannels permeability to different molecules.

Despite the introduction is really good focus and easy to understand, the results section needs a little bit of work on the authors side. I found that the description of the results is not really clear, and the data presentation is confusing most of the times. I suggest the authors to deep read the paper since I found pieces of text that have been moved from where they should be (e.g. from line 370-381).

Response: We thank the reviewer for this comment. This new version of the manuscript was re-organized and re-wrote, particularly, in the result section in order to address the reviewer’s concern about the description and presentation of data. In special, we divided the old Fig.1 into the new Fig. 1 and Fig. 2 and similarly, the old Fig. 2 was transformed in the new Fig. 3. The latter was done with the purpose of giving a clearer idea of our findings. In addition, we amended the location of figure legends that were misplaced throughout the previous version of our manuscript given an error in the word format.

Major comments

*Please show some picture regarding molecules uptake. If the experiments were performed by using time-lapse fluorescence imaging after dye-injection pictures should be shown. Please provide representative pictures for all the dye experiments (Fig 1, Fig 2, Fig 4 and Fig 5).

Response: We thank the reviewer for this comment. In this new version of our manuscript we included representative pictures of dye uptake experiments according to the reviewer´s suggestions. It should be noted that as is mentioned in the method section, image capture of dye coupling experiments was done 3 min after dye injection.

*When Cx43 IF was performed and analyzed by confocal microscopy (Fig 1F), what are the green dots that do not correspond to any cells? Can the author show a better image and lower magnification to fully convince that the green dots correspond to Cx43.

Response: We understand this concern. The image in Fig. 1F correspond to the labeling of both Cx43 and the glial fibrillary acidic protein (GFAP). The latter is an intermediate filament protein usually used as an astrocytic marker, but due its nature does not serve as a reliable indicator of plasma membrane areas. In addition, it should be noted that GFAP is not always expressed homogenously in astrocytes and indeed, there are subpopulations that do not express this marker (PMID: 22144298; PMID: 20012068; PMID: 15145537). In this new version of the manuscript, we improved the quality of old Fig. 1F and showed separately the different channels of Etd, GFAP and Cx43 labeling along with a zoomed inset depicting Cx43 immunoreactive dots in the cell periphery.

*Figure 2. I suggest the authors to re-do this figure since the way it is presented it is really confusing according to what is written in the text. Could you please separate in different graphs the effect of gap19 and Tat-L2, the effect when added TNFalpha and IL1N, and the mice results? Also, revise figure mentioning in the text, I found no explanation for Fig 2 I-L. In this line, do the author consider that graphs shown in Fig 2 E-H are sigmoidal? Please, revise and correct in the text.

 Response: Thank you very much for this comment. We amended the figure and graphs according to the reviewer´s suggestion. In addition, we described properly the old Figures 2 I-L and 2 E-H. We thank the reviewer for noting these errors.

*Please explain whythe authors treat the cells with 70% Ethanol, it seems no clear in the text.

Response: We understand this concern. Ethanol (70%) was used to induce membrane permeabilization of astrocytes and see whether saturation of the nucleic acid binding sites might contribute to the cooperative (S shape) dye uptake observed in our study. Given that 70% ethanol caused a pronounced increase in dye uptake rate to levels far above the maximal values observed with unpermeabilized cells, it can be concluded that the number of intracellular binding sites did not affect the kinetics or saturating uptake rate. In this new version of the manuscript, we included this explanation in the text:

“Ethanol (70%) was used to induce membrane permeabilization of astrocytes and see whether saturation of the nucleic acid binding sites might contribute to the cooperative (S shape) dye uptake observed in our study (Fig. 3Q-T). Given that 70% ethanol caused a pronounced increase in dye uptake rate to levels far above the maximal values observed with unpermeabilized cells, the number of intracellular binding sites likely did not affect the kinetics or saturating uptake rate (Fig. 4A).”

*Figure 5. Same problem as in fig2, data presentation is confusing. Again, could you please separate in different graphs the results corresponding to the inhibitors and mice cells?

Response: Thank you very much for this comment. We amended the figure and graphs according to the reviewer´s suggestion.

*From line 322 to 335, figure reference in the text should be in bold.

Response: Thank you very much for this comment. We amended the references according to the reviewer´s suggestion

*Figure 6. What does * in the pictures means? Is refers to the injected cell? Please provide this information in the text or legend. How was the quantification done? By counting positive cells? I would like to see bright field images showing cell confluence.

Response: Thank you very much for this comment. As inferred by the reviewer, this symbol indicates the injected cell. In this version of the manuscript, we included this statement. As mentioned in the method section, the coupling index was scored as the mean number of cells to which the dye spread from the injected cell to more than one neighboring cell. We now included the bright fields for the dye coupling experiments according to the reviewer´s suggestion.

Reviewer 2 Report

This study has been very well conducted and provides excellent insights into biochemical an pharmacological properties of astrocytic Cx43 hemichannels.

it provides important evidences that were very well explicited by the authors themselves:

-certain stimuli may induce opposite changes in dye uptake kinetics of hemichannels depending on the molecule studied, along with the mismatch between ionic vs. small molecule permeation, raise the need of pursuing new methodological approaches to study these channels.

-this study [provides] the first evidence supporting that an inflammatory condition alters the uptake of cationic molecules via astroglial Cx43 hemichannels depending on properties of the permeant species.

It is quite important to understand how astrocytes, beyond their recently understood major neurophysiological roles, may also contribute to pathogenic mechanisms via altered signalling cascades under inflammatory conditions. In this respect, the choice of IL-1β/TNF-alpha is judicious, as they represent the dominant proinflammatory cytokines secreted by microglia, the innate immune activation of which  is a common point to most neuroinflammatory and neurodegenerative diseases.

This study also provides important guidelines for further studies in the domain, as again well explicited:

"The idea that certain stimuli may induce opposite changes in dye uptake kinetics of hemichannels depending on the molecule studied, along with the mismatch between ionic vs. small molecule permeation, raise the need of pursuing new methodological approaches to study these channels".

This is quite important whn many studies not having the global and multifaceted approach as this one, will provide diverging results, not understanding that they are not showing discrepancies but various effects of the same depending on what the have been analyzing and how.

 To conclude, I agree that this study now calls for future studies on the role of astrocytes and their favorable or worsening behavior under inflammatory conditions of CNS diseases, taking carefully all the information provided by this study, not to generate biased studies claiming misleading conclusions.

i would congratulate the authors for this very good study and its important outcomes.

Author Response

We really appreciate the positive commentaries of the reviewer regarding the scope and impact of our study.

Round 2

Reviewer 1 Report

The authors addressed the main concerns